# Clonal Micropropagation of *Cymbidium erythrostylum* Rolfe

Muthab Hussien [1], Viktoriya Kryuchkova [2,*], Ekaterina Raeva-Bogoslovskaya [2] and Olga Molkanova [2]

1    Institute of Horticulture and Landscape Architecture, Russian State Agrarian University—Moscow Timiryazev Agricultural Academy, 127550 Moscow, Russia
2    Tsitsin Main Botanical Garden of Russian Academy of Science, 127276 Moscow, Russia
*    Correspondence: vkruchkova@mail.ru

**Abstract:** *Cymbidium erythrostylum* Rolfe is one of the most beautiful species of the genus cymbidium which is used as a cut flower or indoor plant. However, it was registered as a rare species even in its original habitat. This study was carried out to develop a protocol for in vitro propagation of *C. erythrostilum*. We used protosomes obtained by the asymbiotic germination of seed on 1/2 of Murashige and Skoog nutrient medium (MS) supplemented with 1 mg/L 2-Isopentenyl adenine (2iP) as objects of study. During the multiplication stage, the number of formed protosomes on a culture medium containing 0.8 mg/L 6-Benzylaminopurine (6-BAP) was higher ($3.8 \pm 0.2$ protosomes). During the subsequent cultivation on 1/2 MS medium supplemented with 2 mg/L 6-BAP and 0.5 mg/L a-naphthaleneacetic acid (NAA), the highest numbers of shoots ($4.00 \pm 0.19$ shoots/plantlet) and leaves ($4.50 \pm 0.14$ leaves/plantlet) were obtained. At the rooting stage under in vitro conditions, the most effective was the use of 1/2 MS nutrient medium with the addition of 0.5 mg/L of indolyl-3-butyric acid IBA, 1 g/L charcoal, and 50 g/L banana puree. The obtained plants were successfully adapted to a substrate consisting of bark, perlite, and peat in a ratio of 1:1:1.

**Keywords:** orchids; propagation; indoor plants; growth regulators; cymbidium; in vitro

## 1. Introduction

The genus *Cymbidium* Swartz belongs to the family Orchidaceae, which includes approximately 50 species. Certain scientists classify plants of this genus as epiphytes, and others as terrestrial orchids. Species and hybrids of this genus are known for their beautiful inflorescences [1,2].

The members of this genus rank first among orchids in the production of cut flowers due to their beautiful inflorescences. They are famous both in flower arrangements and in indoor gardening. Cymbidium is used as traditional medicine for the treatment of inflammation, paralysis, bone fractures, fever, ulcers, and burns [3–5].

*Cymbidium erythrostylum* Rolfe is one of the high ornamental members of the genus *Cymbidium*, which is characterized by early flowering and a unique flower structure. This species was discovered in 1891 by Wilhelm Micholitz and added to the collection of the Royal Botanic Gardens in Glasnevin, where it was formally described by Robert Allan Wolfe in 1905 [6]. It grows in the mountains of Vietnam at an altitude of about 5000 feet above sea level, where it grows not only on the ground, but also a lithophyte and epiphyte [7].

The members of this species are distributed in moist broad-leaved evergreen and deciduous forests along the tops of heavily eroded limestone hills. However, in its original habitat, it has been recorded as a rare species [8]. *Cymbidium erythrostylum* produces large, crystal white flowers with a yellow lip, the tip of which is colored in various shades of pink, purple or red. The petal width is greater than many other species of *Cymbidium*. The flowers are gathered in raised inflorescences of 6–10 units. Leaves are long, linear, about 50 cm long and 1.5 cm wide [9].

Despite the fact that this species is susceptible to botrytis [6], it remains used in breeding programs due to its beautiful flowers. Most commercial hybrids of Cymbidium

are a cross between *C. erythrostylum* with other species such as *C. giganteum*, *C. eburneum*, *C. hookerianum*, *C. sanderae*, *C. lowianum*, *C. tracyanum*, *C. insigne* [10].

Some species of orchids are difficult to propagate by vegetative and generative ways [11]. To maintain the various characteristics of Cymbidium, they are propagated by dividing. Propagation of Cymbidium by separation of pseudobulbs is a slow process. This method is characterized by a low multiplication rate and a long growing cycle (the plant blooms after 3–4 years). This process takes place even more slowly in greenhouses. *Cymbidium* is also very susceptible to viruses, such as the Cymbidium mosaic virus (CymMV; genus *Potexvirus*, family Alphaflexiviridae), which can be spread by means of vegetative propagation methods [12]. Propagation of orchids from seed in nature is limited due to the absence of endosperm and the obligate symbiotic relationship with the mycelium of a certain species of fungus for the germination of small embryos. In addition, during seed propagation, splitting occurs according to the main characteristics of the plant, and varietal purity is not preserved.

All these aspects of orchid propagation have inhibited cut flower production for many years. The only solution to solve such problems is the clonal micropropagation of Cymbidium [13]. Clonal micropropagation is an important instrument both in scientific research and in commercial production. This is an effective way to rapidly obtain virus-free orchid plants due to a high propagation rate, a shortened in the life cycle, the production of planting material during one year [14], higher reproduction rate, and rapid adaptation to ex vitro conditions.

Asymbiotic germination of orchid seeds is often used for the production of valuable species of orchids and has been shown to be an effective way for their preservation [15]. In many studies, it was determined that orchid seed can germinate at high rates and form seedlings on nutrient media such as Knudson and Murashige and Skoog. Subsequently, many protocols were developed for rapid propagation of tropical orchids [16,17].

A characteristic feature of asymbiotic germination of orchid seed is the formation of protocorms containing leaf primordia and primitive stems. Later they develop into seedlings with roots, large leaves, and pseudobulbs [18].

Many researchers distinguish the following stages of orchid development in in vitro culture (Figure 1) [19,20].

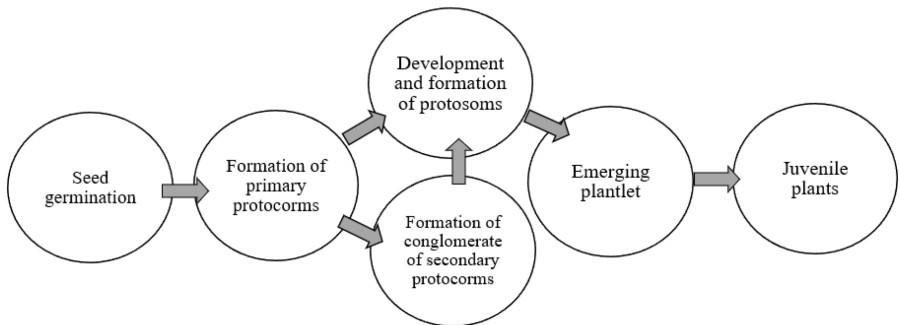

**Figure 1.** Asymbiotic seed germination of orchid in vitro (author M. Hussien).

During the differentiation of protosomes into seedlings, two different types of development were identified:

The protosome consists only of protocorm (the initial stage of seedling development, characteristic of many tropical species).

The protosome contains protocorm and protorizom (a longer stage of seedling development) [21].

This shows the ability of these structures for vegetative propagation and the formation of new seedlings, which can later differentiate into juvenile plants.

In the process of studying the scientific literature, a lot of research was discovered about the effect of growth regulators, mineral compositions of nutrient media, sources of carbohydrates and vitamins, as well as the use of organic additives on the growth,

multiplication factor, and rooting of other species of Cymbidium in in vitro culture. It is worthwhile to note that to date, few studies have been conducted on clonal micropropagation of *C. erythrostylum* [22–24].

This work aims to develop a protocol for in vitro propagation of *Cymbidium erythrostilum*.

## 2. Materials and Methods

The study was carried out in the Laboratory of Plant Biotechnology of the Tsytsin Main Botanical Garden of Academy of Sciences. The objects of study were *C. erythrostylum* protosomes obtained by asymbiotic germination of seed on 1/2 Murashige and Skoog nutrient medium (further MS medium), supplemented with 1.0 mg/L 2-isopentenyladenine (2iP).

At the multiplication stage, the nutrient medium 1/2 strength MS was used with the addition of 1.0 g/L activated charcoal and 6-Benzylaminopurine (further 6-BAP) at various concentrations (0.5, 0.8, and 1.0 mg/L). Hormone-free 1/2 MS medium was used as a control.

At this stage, the number of new protosomes, number of leaves, number of roots, and root length per protosome were evaluated after being cultured for two months. At the stage of growth and development, nutrient medium 1/2 MS was used, supplemented with different cytokines, Kinetin and 6-BAP, at concentrations of 1.0 and 2.0 mg/L. Each variant was supplemented with 0.5 mg/L a-naphthaleneacetic acid (further NAA) and 1.0 g/L activated charcoal. At this stage, the following parameters were assessed: seedling length, number of shoots, number of leaves, number of roots, and root length.

Plantlets were rooted on a nutrient medium 1/2 MS containing 0.5 mg/L indolyl-3-butyric acid (further IBA) with the addition of various organic additives (100 mL/l coconut water, 20 g/L sweet potatoes, 50 g/L banana puree). As a control, 1/2 MS nutrient medium without organic additives was used. The experiments were carried out in 5 replicates, each replicate contained approximately 7 explants. The culture room was set at 25 ± 2 °C, 70 ± 5% relative humidity with light intensity of 1500 to 2000 Lux with a 16/8 h (light/dark).

The rooted seedlings were transferred from the nutrient medium and washed under running tap water. Then they were placed onto plastic containers containing equal proportions of peat, perlite and bark, and the plantlets were sprayed with water twice weekly. The plantlets were kept in the greenhouse for further acclimatization. At this stage, the survival rate was calculated after 3 months.

The results of the experimental data were processed statistically using the computer programs of Microsoft Office Excel 2016 and PAST 2.17c. When determining the significant difference between the experimental variants, the least significant difference (LSD) at a level of $p < 0.05$ was used.

## 3. Results

At the first stage of research, the main task was to obtain the largest number of protosomes. The results of study showed that the formation of new protosomes depended on the concentration of 6-BAP (Figure 2).

The nutrient medium 1/2 MS containing 0.8 mg/L of 6-BAP showed a greater number of formed protosomes (3.8 ± 0.2 units) compared with other treatments. The minimum number of protosomes was formed on the control variant without the addition of 6-BAP.

Based on the comparison, the difference between group averages and least significant difference, the number of formed protosomes significantly differs from the control at a 5% significance level in variants with concentrations of 0.5 and 0.8 mg/L of 6-BAP. At a concentration of 1.0 mg/L, the number of protosomes does not statistically differ from the control (Figure 3).

At the same time, on the same nutrient medium, before transplantation, *C. erythrostylum* protosomes formed leaves and roots. At this stage, no significant differences were detected in their number depending on the concentration of 6-BAP (Table 1).

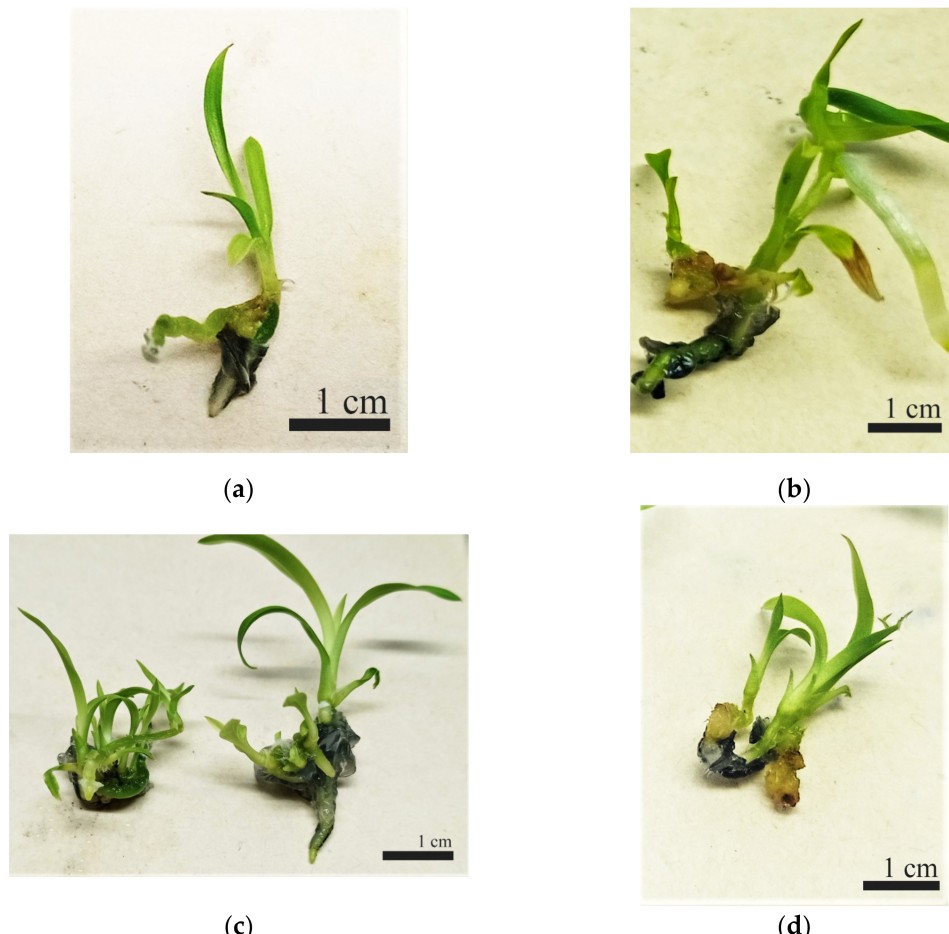

**Figure 2.** Influence of different concentrations of 6-BAP on the formation of new protosomes: (**a**) control; (**b**) 0.5 mg/L; (**c**) 0.8 mg/L; (**d**) 1.0 mg/L (Bars = 1.0 cm).

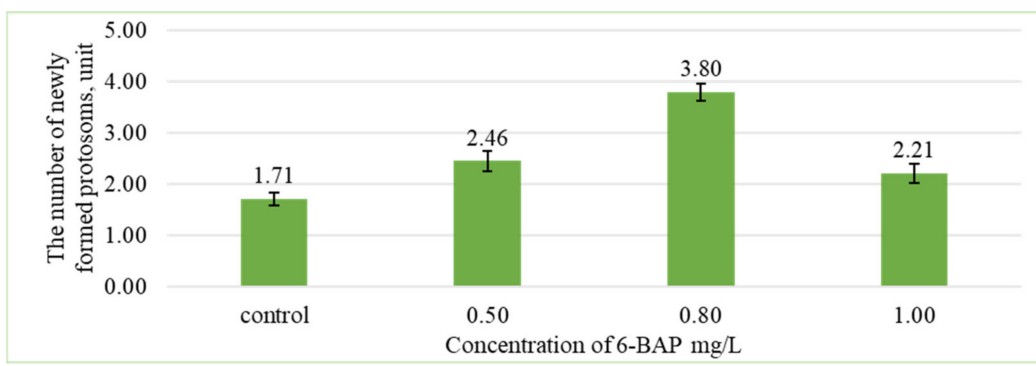

**Figure 3.** Influence of different concentrations of 6-BAP on the number of newly formed protosomes after 60 days of cultivation ($LSD_{05}$ 0.68).

**Table 1.** Influence of 6-BAP concentration on the morphometric parameters of *C. erythrostylum* protosomes after 60 days of cultivation.

| Concentration of 6-BAP mg/L | Number of Leaves, Unit | Number of Roots, Unit | Root Length, cm |
|---|---|---|---|
| Control | 2.22 ± 0.10 * | 1.14 ± 0.10 | 0.48 ± 0.11 |
| 0.5 | 2.89 ± 0.21 | 1.14 ± 0.10 | 0.76 ± 0.13 |
| 0.8 | 2.61 ± 0.23 | 1.07 ± 0.07 | 0.60 ± 0.13 |
| 1.0 | 2.83 ± 0.22 | 1.14 ± 0.18 | 0.73 ± 0.14 |

* Values are mean ± SD.

The number of leaves at the stage of protosome formation was satisfactory, but the number and length of the roots were small. To ensure a more balanced development, auxins were added to the nutrient medium in addition to cytokinins at the next stage of reproduction. It is known that when combining cytokinins and auxins, it is recommended to use higher concentrations of cytokinins. In this regard, cytokinin concentrations of 1.0 and 2.0 mg/L were used.

In the present study, the influence of different concentrations of auxins and cytokinins on the morphometric parameters of *C. erythrostylum* explants was established (Table 2; Figure 4). The most important parameters are the number of shoots, the number of leaves and the number of roots, as well as the height of the plant and the length of the roots.

**Table 2.** Influence of growth regulators on the morphometric parameters of *C. erythrostylum* plantlets after 120 days of cultivation.

| Concentration of Growth Regulators, mg/L | Plant Height, cm | Number of Shoots, Unit | Number of Leaves, Unit | Number of Roots, Unit | Root Length, cm |
|---|---|---|---|---|---|
| 1.0 6-BAP + 0.5 NAA | $2.22 \pm 0.20$ a * | $1.86 \pm 0.23$ a | $2.59 \pm 0.10$ a | $1.31 \pm 0.15$ a | $1.00 \pm 0.08$ ab |
| 2.0 6-BAP + 0.5 NAA | $3.19 \pm 0.17$ b | $4.00 \pm 0.19$ c | $4.50 \pm 0.14$ c | $1.93 \pm 0.11$ b | $1.59 \pm 0.12$ b |
| 1.0 Kin + 0.5 NAA | $3.00 \pm 0.20$ b | $2.80 \pm 0.22$ b | $3.18 \pm 0.11$ b | $1.87 \pm 0.17$ b | $1.31 \pm 0.14$ b |
| 2.0 Kin + 0.5 NAA | $2.09 \pm 0.16$ a | $1.80 \pm 0.17$ a | $2.27 \pm 0.19$ a | $1.37 \pm 0.12$ a | $0.83 \pm 0.04$ a |
| LSD$_{05}$ | 0.59 | 0.99 | 0.45 | 0.43 | 0.37 |

* Values are mean $\pm$ SD, the letters "a", "b" and "c" denote groups that significantly differ from each other at a 5% level of significance based on the Tukey criterion.

The results showed that the highest number of shoots ($4.00 \pm 0.19$ units/explant) and the of leaves ($4.50 \pm 0.14$ units/explant), significantly different from other variants, was noted when using 2.0 mg/L 6-BAP in combination with 0.5 mg/L NAA (Figure 5).

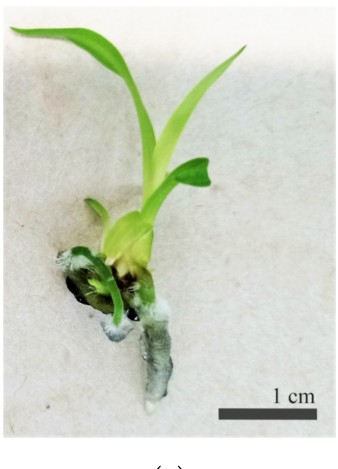

(**a**)

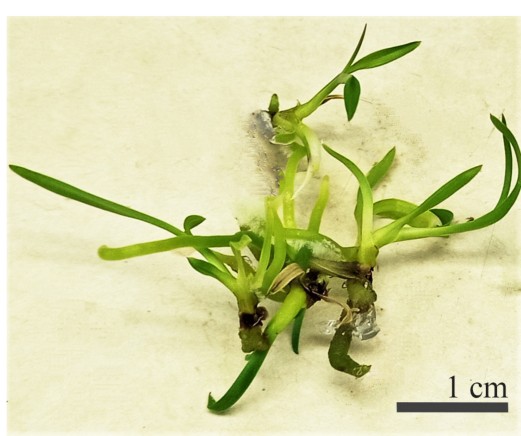

(**b**)

**Figure 4.** *Cont*.

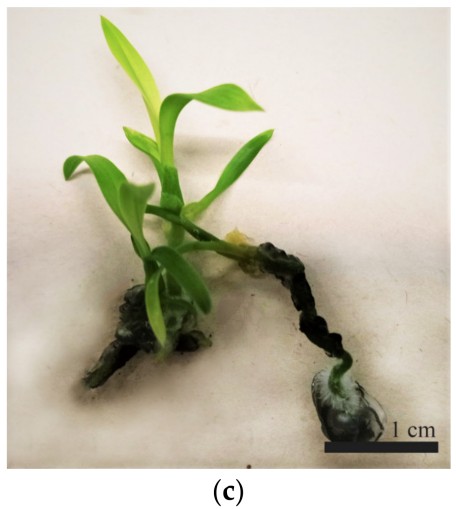

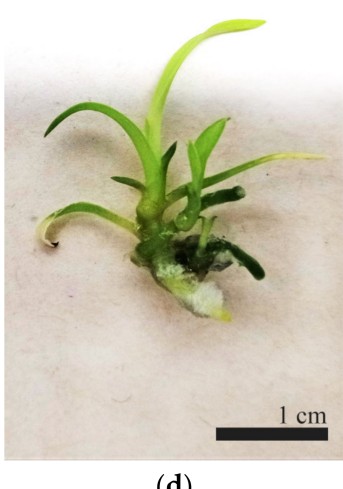

(**c**)                                                                              (**d**)

**Figure 4.** Influence of growth regulators on the morphometric parameters of *C. erythrostylum* plantlets: (**a**) 1.0 6-BAP + 0.5 NAA mg/L; (**b**) 2.0 6-BAP + 0.5 NAA mg/L; (**c**) 1.0 Kin + 0,5 NAA mg/L; (**d**) 2.0 Kin + 0.5 NAA mg/L.

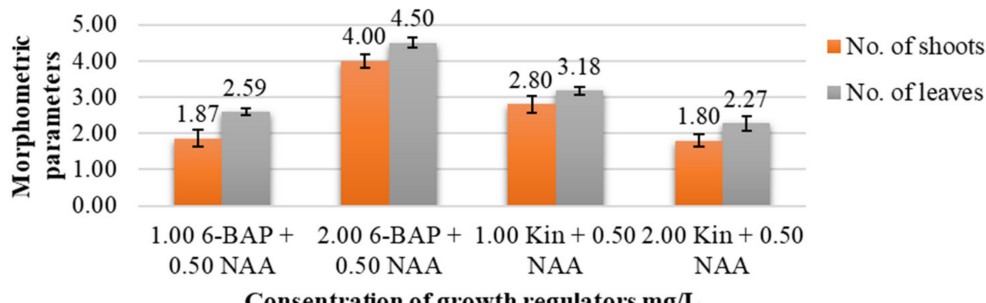

**Figure 5.** Effect of different growth regulators on shoot and leaf number of *C. erythrostylum* plantlets at 120 days of in vitro growth ($LSD_{05}$ for number of shoots = 0.99; $LSD_{05}$ for number of leaves = 0.45).

During subsequent cultivation on 1/2 MS nutrient media containing 2 mg/L 6-BAP + 0.5 mg/L NAA; 1 mg/L Kin + 0.5 mg/L NAA), an increase in plant height ($3.19 \pm 0.17$ cm; $3.00 \pm 0.20$ cm, respectively) was detected compared with other treatments. At the same time, the length and number of roots increased on all nutrient media, but no significant differences were detected between the variants (Table 2).

By all parameters, the optimal nutrient medium for cultivating plants *C. erythrostylum* is 2.0 mg/L 6-BAP + 0.5 mg/L NAA, the worst indicators are noted on 2.0 mg/L Kin + 0.5 mg/L NAA medium.

At the rooting stage, the 1/2 MS nutrient medium containing 0.5 mg/L IBA, 1 g/L charcoal, and 50 g/L banana puree showed a significant difference for number of roots; at the same time, it was observed that the longer roots formed on a nutrient medium containing 100 mL/L coconut water (Table 3, Figure 6). These options are significantly different from the rest.

**Table 3.** Effect of organic additives on in vitro rooting of *C. erythrostylum* plantlets at 180 days of in vitro growth.

| Organic Additives | Number of Roots, Unit | Root Length, cm |
|---|---|---|
| Control | 1.69 ± 0.11 a * | 1.21 ± 0.10 a |
| coconut water | 2.24 ± 0.14 b | 3.53 ± 0.15 c |
| banana puree | 3.93 ± 0.20 c | 1.72 ± 0.48 b |
| sweet potatoes | 2.86 ± 0.12 b | 2.14 ± 0.10 b |
| LSD$_{05}$ | 0.54 | 0.44 |

* Values are mean ± SD, the letters "a", "b" and "c" denote groups that significantly differ from each other at a 5% level of significance based on the Tukey criterion.

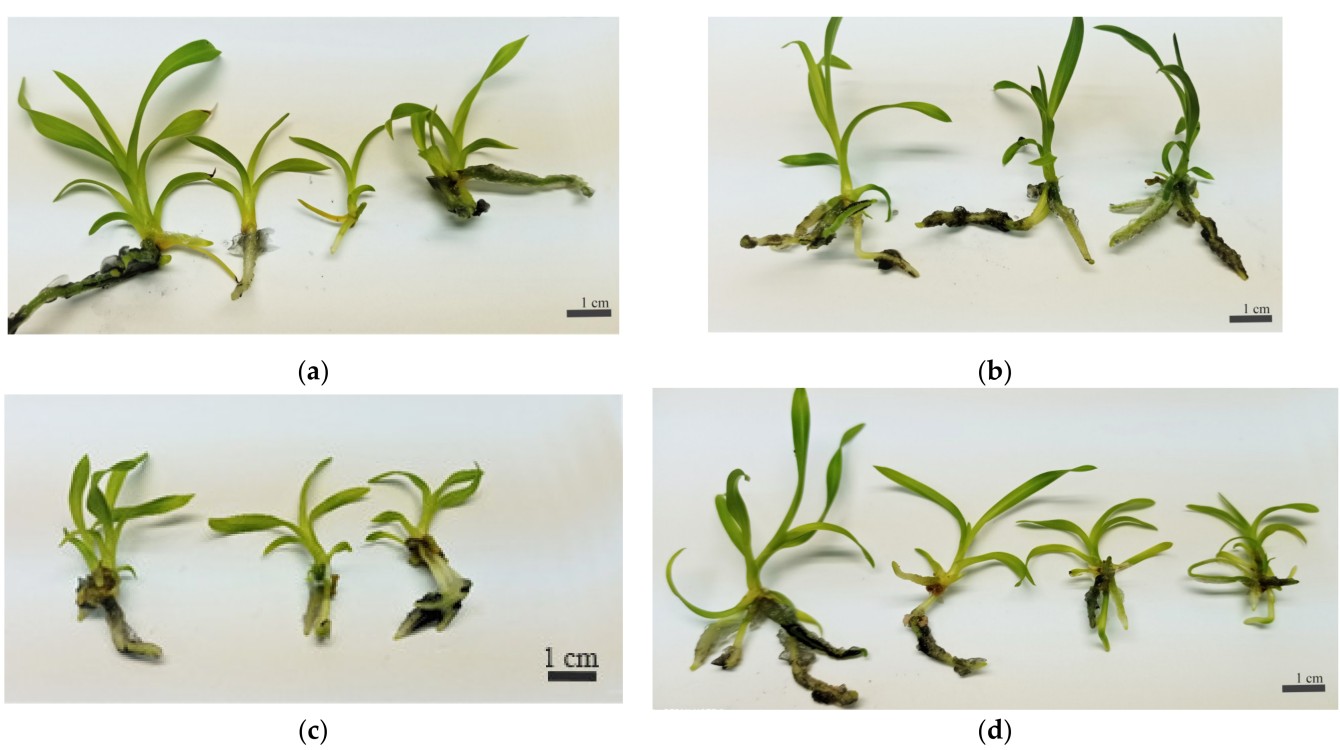

(**a**)  (**b**)

(**c**)  (**d**)

**Figure 6.** Effect of organic supplements on root parameters of *C. erythrostylum* seedlings: (**a**) 0.5 mg/L IBA (control); (**b**) 0.5 mg/L IBA + banana puree; (**c**) 0.5 mg/L IBA + coconut water; (**d**) 0.5 mg/L IBA + sweet potato.

Under ex vitro conditions, the well-rooted plants were adapted successfully in a substrate consisting of bark, perlite, and peat in the ratio of 1:1:1, respectively. The survival rate of plantlets transferred to the greenhouse after 90 days was 100% (Figure 7).

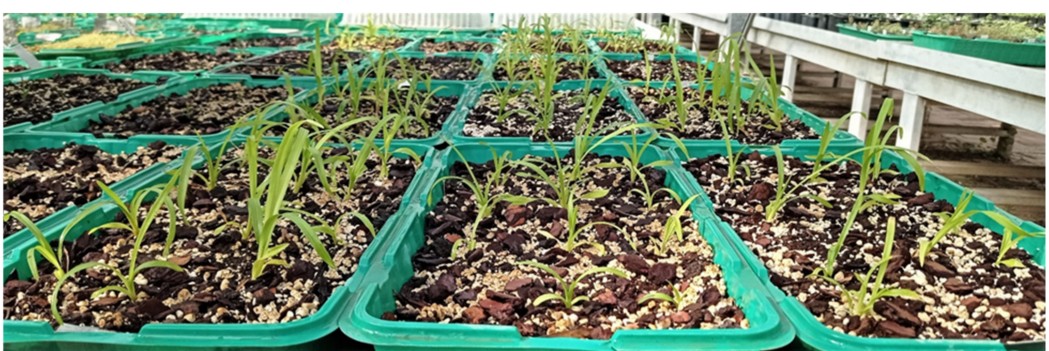

**Figure 7.** Plantlets of *C. erythrostylum* growing in containers in greenhouse.

## 4. Discussion

Each plant species has certain nutrient requirements for propagation, appearance and development of shoots and leaves, and root formation under in vitro conditions [25]. The process of formation of new protosomes is an important process for clonal micropropagation of orchids [26]. In the first experiment, our goal was to obtain the maximum number of protosomes. The number of formed protosomes on a nutrient medium containing 0.8 mg/L of 6-BAP was maximum ($3.8 \pm 0.2$ units) compared with other variants. This is consistent with the results of other studies, in which they reported the effect of 6-BAP on the propagation, formation, and development of new shoots in members of the genus *Cymbidium* such as *C.giganteum* Wall. ex Lindl, *C. aloifolium* (L.) Sw., *C. faberi* Rolfe, *C. lowianum* Rchb., *C. finlaysonianum* Lindl, *Cymbidium* "Sleeping Nymph" [3,27–29]. In addition, neither too low nor too high concentrations of cytokinins were effective for obtaining a high multiplication rate, and high concentrations could cause variations in plants.

However, Trunjaruen and Taratima (2018) [26], Peres et al. (1999) [30], and other authors noted the inhibitory effect of cytokinin on the growth and development of roots of *C. aloifolium* and other orchid species. These results are similar to those obtained in this study when explants were grown on media containing only cytokinins as growth regulators.

Many studies have reported the effectiveness of using cytokinin in a nutrient medium in combination with auxin for the formation and subsequent growth of shoots [13,14,26]. This is consistent with the results of our study, where the highest numbers of leaves and shoots were observed on nutrient medium that contained 2 mg/L 6-BAP and 0.5 mg/L NAA. This can be explained by the fact that the combination of NAA and 6-BAP is the most appropriate choice for clonal micropropagation of other *Cymbidium* species, not only because they are cheaper than other growth regulators, but also because their active ingredients are not completely destroyed at high temperature (121 °C) [13,31].

In some research papers, it has been indicated that the combination of auxin and cytokinin can increase plant height in many species of orchid [26,32]. In our studies, it was found that during subsequent cultivation on 1/2 MS nutrient media containing 2 mg/L 6-BAP + 0.5 mg/L NAA; 1 mg/L Kin + 0.5 mg/L NAA, an increase occurred in plant height compared to other variants of nutrient media. At the same time, the length and number of roots increased on all nutrient media, but no significant differences were detected between the variants. This can be explained by the fact that the use of cytokinins in high concentrations together with auxin can lead to a reduced effect of auxin on root formation [33].

The nutrition medium for growth and development is not suitable for the formation of roots in young seedlings. Consequently, the formed seedlings were cultivated on a nutrient medium for rooting in order to create a strong root system. At the multiplication stage, the main task is to obtain more shoots and well-developed leaves, and the formation of roots is secondary. However, the presence of roots at this stage increases the indicators of rhizogenesis in the future.

Seedlings on a 1/2 MS nutrient medium supplemented with 0.5 mg/L IBA and banana puree formed more numerous and long roots that were still thick and hairy compared to other variants; however, the addition of sweet potatoes or coconut water to the nutrient medium provided good results compared to control. It was observed that long and thin roots formed on a nutrient medium containing coconut water.

The survival rate of plantlets ex vitro strongly depends on the formation of a powerful and healthy root system in vitro. Many studies have reported the effectiveness of using IBA in low concentrations for rooting different species of *Cymbidium* [13,14].

It was recommended to use organic additives in the nutrient medium at the rooting stage of representatives of orchids because of their high sugar content, organic acids, amino acids, a lot of vitamins, phytohormones (cytokinins and auxins), and minerals that contribute to the formation of roots [34,35].

Kaur and Bhutani (2012) [36], Fang et al. (2011) [37], and other authors noted the stimulating effect of banana puree on the formation of a strong root system for many Cymbidium species under in vitro conditions.

They also recommended using nutrient media containing auxins and banana puree, as it contains large amounts of potassium, calcium, sodium, zinc, riboflavin, niacin, ascorbic acid, peroxin A, and many growth regulators such as gibberellin, zeatin, and IAA, in addition to the high content of carbohydrates that provide energy to seedlings in vitro [38].

The final stage of clonal micropropagation is the adaptation of seedlings under ex vitro conditions. The ability of seedlings to tolerate stress during transplantation often determines the success or failure of the clonal micropropagation process in in vitro culture [39]. The choice of a suitable substrate with high aeration, permeability, and an appropriate acidity grade is a prerequisite for ensuring autotrophic growth [40]. The most used substrates for the adaptation of orchids consist of bricks or charcoal, clay tiles, bark, coconut fibers, sawdust, perlite, vermiculite, peat, or sphagnum moss in different ratios [41–43]. This is consistent with our results, where the seedlings were adapted successfully in a substrate containing bark, perlite, and peat in the ratio of 1:1:1, respectively.

### 5. Conclusions

The results of the present investigation showed that during the multiplication stage, among different concentrations of 6-BAP, 0.8 mg/L 6-BAP provides a higher number of formed protosomes in *C. erythrostylum*. In a combined effect of growth regulators on plant regeneration, a nutrient medium containing, 2 mg/L 6-BAP and 0.5 mg/L NAA provide better results for plant growth and development. At the rooting stage, the results revealed that adding 0.5 mg/L of IBA, 1 g/L charcoal, and 50 g/L banana puree to the 1/2 MS medium was the most influential for root induction. Juvenile plants can be successfully adapted and grown in a substrate (bark, perlite, and peat) in a ratio of 1:1:1. The results of this study allow the establishment of an efficient protocol for in vitro propagation of *C. erythrostylum*, which may be applicable for large-scale production. In the future, this species must be studied in the greenhouse and suitable conditions must be found to reach the flowering stage.

**Author Contributions:** Conceptualization, O.M. and V.K.; methodology, O.M.; validation, O.M. and V.K.; formal analysis, M.H.; resources, O.M. and M.H.; data curation, M.H. and V.K.; writing—original draft preparation, M.H.; writing—review and editing, O.M. and V.K.; visualization, M.H. and E.R.-B.; supervision, O.M. All authors have read and agreed to the published version of the manuscript.

**Funding:** The reported study was supported by assignments 122042700002-6 of the Ministry of Science and Higher Education of the Russian Federation.

**Institutional Review Board Statement:** Not applicable.

**Informed Consent Statement:** Not applicable.

**Data Availability Statement:** Not applicable.

**Conflicts of Interest:** The authors declare no conflict of interest.

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
