# Peer review of "Clonal Micropropagation of Cymbidium erythrostylum Rolfe"

_2037-0164, doi:10.3390/ijpb14010003_

Round 1

Reviewer 1 Report

The research article: “Clonal micropropagation of Cymbidium erythrostylum Rolfe” contains original results of investigations carried out by the  authors.

Overall, the work is well planned and presented. The title of the work and the abstract fully reflect the content of the article. The introduction contains brief information that will help the reader to understand this issue. Materials and methods were described in detail. But, unfortunately, all the results obtained are not well substantiated and are being discussed.

1.    According to the research results, there is no justification in the article why exactly such concentrations were used: the effect of the concentration of growth regulators - BA 0.5; 0.8 and 1 mg/l. But according to the data obtained at these concentrations, no significant differences were found in the amount of formation from protosomes of leaves and roots.

2.    In Table 2, a new medium of 2 mg/l BA and 0.5 mg/l NAA appears (the best option, morphometric indicators were measured) in contrast to the previously used 2 mg/l kinetin and 0.5 mg/l NAA. There is also no explanation why this environment was used. Just on the medium of 2 mg/l BA and 0.5 mg/l NAA, good data on the formation of leaves and shoots are presented. But according to the data obtained, it turns out that there are still no significant differences.

3.    When rooting, the authors used high concentrations of cytokinins, which affected poor rooting. It is known that at the rooting stage it is desirable to reduce the concentration of cytokinins and increase the concentration of auxins.

4.    The authors write that the obtained plants have been successfully adapted to ex vitro conditions. But there is no data.

5.    The discussion is very weak.

In general, after correcting the comments, the manuscript can be published.

Author Response

Thank you for carefully reviewing our work. The text has been corrected. Please view the attached files.

Point 1: According to the research results, there is no justification in the article why exactly such concentrations were used: the effect of the concentration of growth regulators - BA 0.5; 0.8 and 1 mg/l. But according to the data obtained at these concentrations, no significant differences were found in the amount of formation from protosomes of leaves and roots.

 Response 1: At the first stage, the main task was to obtain the maximum number of protosomes, cytokinin concentrations were selected based on literature data, since lower concentrations do not lead to mass formation of protosomes, and higher concentrations can lead to spontaneous formation of somaclonal variants [25, 27, 29]. Since we needed to get protosomes, pure cytokinins were used. However, such a nutrient medium is not suitable for further cultivation. But it is possible to focus on the concentration of cytokinins from the first experience at the following stages of cultivation

Point 2: In Table 2, a new medium of 2 mg/l BA and 0.5 mg/l NAA appears (the best option, morphometric indicators were measured) in contrast to the previously used 2 mg/l kinetin and 0.5 mg/l NAA. There is also no explanation why this environment was used. Just on the medium of 2 mg/l BA and 0.5 mg/l NAA, good data on the formation of leaves and shoots are presented. But according to the data obtained, it turns out that there are still no significant differences.

Response 2: Table 2 shows the results obtained at the multiplication stage, the purpose of which is to obtain the maximum number of shoots. According to the indicator of the number of shoots and the number of leaves, the optimal cultivation option is determined. In this case, the number and length of the roots is of auxiliary importance, although it is important for subsequent stages. the concentrations of growth regulators were selected based on the results of previous experience and literature data that the addition of auxins reduces the effect of cytokinins [13, 14].

 Point 3: When rooting, the authors used high concentrations of cytokinins, which affected poor rooting. It is known that at the rooting stage it is desirable to reduce the concentration of cytokinins and increase the concentration of auxins.

 Response 3: For rooting, we used a 1/2 MS nutrient medium with the addition of 0.5 mg/l IBA, 1 g / l of charcoal and additional organic additives. The results are presented in Table 3. In this experiment, there are no cytokinins in the nutrient medium, good results were obtained in the number and length of roots.

 Point 4: The authors write that the obtained plants have been successfully adapted to ex vitro conditions. But there is no data.

 Response 4: The results obtained in the process of plant adaptation have been added to the section "research results". In our studies, all plants adapted well to ex vitro conditions

 Point 5: The discussion is very weak.

 Response 5: The discussion is expanded

Reviewer 2 Report

This interesting article describes the micropropagation of Cymbidium erythrostylum Rolfe, an orchid species. Different phytohormones and organic additives were used for optimization. The appropriate selection of these allowed for optimization of the number of protosomes as well as root induction and, finally, to ensure the growth of juvenile plants.

The article is well written understandably and the selection of a rarer ornamental plant (instead of the otherwise mostly described food or secondary metabolite-producing plant species) appeals to me very much.

Unfortunately, the article cannot yet be published in its current form, as some aspects need to be further elaborated. In addition, the article needs to be formally revised because the number of stylistic errors and inconsistencies is too high. I, therefore, recommend that the article be revised. My main points that led to this decision are listed below:

 Comments:

1.       The introduction discusses the problems of traditional propagation of Cymbidium that limit production (lines 49 to 61). In the following paragraphs, the advantages of micropropagation are discussed, and the potential problems of plantlets are not neglected (line 221). However, I lack an objective comparison between the two approaches: What are the merits of micropropagation (time-wise, concerning survival rates, and cost-wise)? Especially the conclusion, which is rather short with 8 lines (lines 246 to 253) and primarily a summary, does not convince me at all. What are the larger implications, why is this approach important and useful, and what can be done with it? Please expand on this a bit more.

2.       The article seems a bit short to me (5 pages of text without tables and graphs) and seems more like a communication than a full article. Why don't you describe some aspects more intensively?

For example, I would be interested in the adaptation to ex vitro conditions:

a.       which is not described at all in the methods,

b.       is dealt with in only one sentence in the results (lines 178 and 179),

c.       and which is addressed in the discussion when selecting the substrate (lines 239 to 244), but aspects such as survival rate as a function of the root system are not considered (addressed in lines 221 to 222). What was the survival rate? What were the differences concerning different root systems, etc.?

3.       The text in its present form contains too many formal errors and inconsistencies, which, overall, could leave a careless impression on the reader. The following are some examples (The list is not exhaustive):

a.       Missing spaces, e.g. “C.erythrostilum” on line 11

b.       Superfluous spaces, e.g. “0.8 mg / L” on line 14 (but without spaces on line 18)

c.       Utilization of “l” and “L” for liter (e.g. “0.8 mg / L” on line 14 and “0.5 mg/l” on line 15). Be consistent.

d.       Inconsistent introduction of abbreviations, e.g. “2iP” on line 12 but no introduction of 6-BAP (line 15), NAA (line 15) and MS (written in full on line 11, abbreviation on line 15).

e.       Missing dots, e.g. “etc [64]” on line 64 or “time It” on line 170.

f.        Part of a paragraph mark in Figure 1.

g.       Utilization of “,” and “.” as decimal separators, e.g. in figure 3 and table 1. Stick to “.”

h.       Missing standard deviations in Figure 3

i.         Typos, e.g. “6-BAR” on line 145

j.         Inconsistent use of significant digits, e.g. “2,8” vs “4,00” in table 2. And, again, use “.” as decimal separator

k.       Missing scale in Figure 6 c)

l.         Not all species names are in italics, e.g. C. erythrostylum” on line 248

m.     Inconsistent reference list, e.g. “DOI:” on line 273, “DOI” on line 285 and “doi:” on line 292

Please revise your manuscript thoroughly once again to add more depth to the exciting subject matter and eliminate the multitude of errors and inaccuracies. I look forward to evaluating the revised version of this interesting article and wish you success in correcting it.

Author Response

Thank you for carefully reviewing our work. The text has been corrected. Please view the attached files.

Point 1: The introduction discusses the problems of traditional propagation of Cymbidium that limit production (lines 49 to 61). In the following paragraphs, the advantages of micropropagation are discussed, and the potential problems of plantlets are not neglected (line 221). However, I lack an objective comparison between the two approaches: What are the merits of micropropagation (time-wise, concerning survival rates, and cost-wise)? Especially the conclusion, which is rather short with 8 lines (lines 246 to 253) and primarily a summary, does not convince me at all. What are the larger implications, why is this approach important and useful, and what can be done with it? Please expand on this a bit more.

 Response 1: The introduction section has been updated. Updated conclusion. The results of this study allow the establishment of an efficient protocol for in vitro propagation of C. erythrostylum, which may be applicable for large-scale production. In the future, this species must be studied in the greenhouse and found the suitable conditions to reach the flowering stage.

Point 2: The article seems a bit short to me (5 pages of text without tables and graphs) and seems more like a communication than a full article. Why don't you describe some aspects more intensively?

Response 2: Agree with the remark. The introduction section has been expanded, results and photos of plant adaptation to ex vitro conditions have been added.

 Point 3: The text in its present form contains too many formal errors and inconsistencies, which, overall, could leave a careless impression on the reader. The following are some examples (The list is not exhaustive):

Response 3: Thanks for the comment, errors and inaccuracies have been corrected

Reviewer 3 Report

The work is of considerable interest, especially due to that the multiplication of a rare and, at the same time, highly commercially interesting species is being analysed.

The introduction correctly clarifies the state of the art, providing the right basis for the research work

The applied methodology is clearly described. It may be appropriate to indicate the light intensity of the culture room

Results. In the tables of the results, indicating the statistically significant differences using letters can clarify the reading. Tab 1 first column, Tab 2, Tab 3 and Fig 5, replace comma with point. Row 145 replace BAR with BAP. Also, it would be interesting if the rate of adaptation to transplantation was reported.

The discussion is well articulated and compares the results with those of previous works. For an easier comparison with the bibliography, the bibliographic references should be indicated with the indication of the number, as in the rest of the text: for example Trunjaruen and Taratima [26] (row192). This check must be done for the whole text, from the beginning.

Author Response

Thank you for carefully reviewing our work. The text has been corrected. Please view the attached files.

Point 1: It may be appropriate to indicate the light intensity of the culture room

 Response 1: Thanks for the comment. The culture room was set at 25 ± 2 °C, 70 ± 5% relative humidity with light intensity of 1500 to 2000 Lux with a 16/8 h (light/dark), Information added to the article

Point 2: Results. In the tables of the results, indicating the statistically significant differences using letters can clarify the reading. Tab 1 first column, Tab 2, Tab 3 and Fig 5, replace comma with point. Row 145 replace BAR with BAP. Also, it would be interesting if the rate of adaptation to transplantation was reported

Response 2: Statistical differences using letters have been added to the result tables. Inaccuracies have been corrected in the text of the article.

 Point 3: The discussion is well articulated and compares the results with those of previous works. For an easier comparison with the bibliography, the bibliographic references should be indicated with the indication of the number, as in the rest of the text: for example Trunjaruen and Taratima [26] (row192). This check must be done for the whole text, from the beginning.

 Response 3: Сorrected

Round 2

Reviewer 2 Report

Thanks for the revision of the article, which reads much better now. Especially the new data on ex vitro cultivation is fascinating, a survival rate of 100% after 90 days is impressive.

In terms of content, the article is thus fine for me. Although many spelling and consistency errors have been corrected, there are still some inaccuracies in the article. I, therefore, advise the authors in their interest to recheck the text. Some examples:

·       Space between units

o    line 14 “mg / L”

o   Line 299, 2 x “mg / L”

·       Missing space between number and unit

o   line 12 “1mg/l”

o   line 19 “1g/l”

o   line 203 “1g/l”

o   line 300 “1g/l”

·       No uniformity for liter, e.g.

o   line 14 “mg / L”

o   line 102 “mg/L”

·       No uniformity for one half (1/2 vs ½)

·       Figure 5 should be Figure 7 (line 223)

·       Missing citations on line 278 (compare to your changes on line 238)